# Study on the Changes in Shielding Performance Based on Electrospinning Pattern Shapes in the Manufacturing Process of Polymer-Metal Composite Radiation Shielding Materials

**Seon-Chil Kim**

Department of Biomedical Engineering, Keimyung University, 1095 Dalgubeol-Daero, Daegu 42601, Republic of Korea; chil@kmu.ac.kr; Tel.: +82-10-4803-7773

**Abstract:** X-rays are commonly employed in medical institutions for diagnostic examinations, which often results in radiation exposure for both patients and medical personnel during treatments and procedures. Hands are typically the most exposed body parts, with scattered rays causing secondary exposure. To address this issue, a lightweight functional radiation protection fabric that guarantees the activity of medical personnel is required. In this study, a shielding fabric was fabricated with nanofibers using a mixture of tungsten and polyurethane to resolve the weight reduction problem of such nanofibers. To improve the shielding performance, the change in the performance arising from the spinning pattern in the nanofiber electrospinning manufacturing process was compared and tracked. The patterns reproduced via electrospinning included honeycomb, matrix-orthogonal, double-circle, and spider web patterns. Through this, a nanofiber fabric was produced, and the shielding performance was evaluated. The honeycomb pattern fabric exhibited the best shielding rate of 89.21% at an effective X-ray energy of 60.3 keV, and the double-circle pattern exhibited the lowest shielding rate of 62.55% at the same energy. Therefore, it was observed that the pattern arising from the nanofiber spinning conditions affects the dispersion of the shielding material, which affects the shielding performance. When 0.3 mm tungsten nanofiber fabric is compared with its lead equivalent of 0.25 mm, a difference of 8.7% was observed, suggesting that the nanofiber can be used in medical institutions. Future research will explore the potential of protective fabrics that minimally impact medical personnel's mobility but provide enhanced protection against radiation exposure.

**Keywords:** radiation shielding; electrospinning; radiation exposure; interventional procedures; tungsten

## 1. Introduction

Radiation-related workers in medical institutions can be directly or indirectly exposed to radiation during the examination and treatment processes; therefore, it is necessary to develop tools that can easily block some of this radiation [1]. For optimal functionality, shielding garments used in healthcare settings should be lightweight, thin, and flexible, allowing medical personnel to move freely. The majority of radiation shielding sheets or fibers are constructed from a composite structure that combines metal particles with high-molecular-weight materials, ensuring effective shielding performance [2,3].

In addition, since metal particles have higher shielding efficiency than polymer materials, they play an active role in radiation blocking [4,5]. Therefore, a technique for evenly dispersing the particles of the shielding material is required to increase the radiation shielding efficiency [6,7]. The dispersion of metal particles in polymer materials is irregular and difficult to predict, and it is difficult to standardize and apply the same dispersion technology in the particle manufacturing process [8,9]. Therefore, in most manufacturing, the degree of dispersion can be predicted by the ratio of the mass of metal particles to the total mass. The same mass ratio can be explained on the basis that it generally provides the

same shielding factor [10]. This method cannot be expected to fundamentally lower the thickness or weight of the shielding sheet. Therefore, in this study, a more effective metal particle dispersion method is proposed to improve the shielding performance. Tungsten, which is used as a shielding material, has an atomic number of 74 and a high density of 19.25 g/m$^3$; therefore, it satisfies the requirements for replacing lead [11].

Polyurethane is used as the polymer material; it is mainly composed of synthetic fibers, has strong and lightweight characteristics with a foam structure, a molecular weight of 100,000 or more, and a density of 0.03 g/cm$^3$ [12]. The bonding form of the polymer and the metal particle was presented as a pattern and the correlation with the shielding performance was analyzed to establish a modelling that can be evenly dispersed in the polymer material.

Conventionally, it is common to improve dispersion by gradually increasing the mixing density through stirring to inject metal particles into a polymer material [13]. Evenly dispersing the metal particles distributes the metal particles in the polymer material, and this is performed in the mixing process. However, the distribution cannot be quantitatively controlled in this process. Therefore, a method for adjusting the mixing ratio in the process or the input and output per hour in the injection process is used. Because the temperature and stirring time of the polymer affect the dispersion of metal particles in the process of mixing shielding materials, the polymer binding method has recently been applied to nanocomposite technology [14,15]. It is possible to implement a more stable dispersion than conventional methods by mixing the particles of the shielding material between the polymer nanofibers. Nanofibers refer to fibers with a diameter of 100 nm or less; these have a high surface area ratio to volume, but are porous, making it difficult to expect a shielding effect [16,17]. It is expected that the nanofibers pores can be minimized depending on which pattern the polymer and metal particles adopt. Therefore, a processing model capable of dispersing micro-particles of tungsten can be proposed when designing a nanofiber pattern by electrospinning. A new process technology is needed to realize a larger amount of metal particles through the process of dispersing metal particles into nanofibers [18]. This method may be described as an electrospinning pattern forming process. A structure in which tungsten particles are well dispersed in the nanofiber strands is required. Therefore, a pattern can be formed with a structure in which metal particles are well dispersed through ideal bonding of polymer chains. This pattern formation dispersion method can enhance the shielding effect by widening the interaction with the incident radiation, and the scattered radiation protection effect of medical institutions can be expected [19]. If nanofibers have limitations in shielding direct rays, lightweight shielding fabrics for shielding scattered rays are flexible; thus, functional radiation protection effects that can guarantee the activity of medical staff can be expected [20]. Therefore, the low-dose shielding effect is verified by using the pattern of nanofibers produced in this study, and simultaneously, a comparative evaluation of standard lead is also performed to verify commercialization. In this study, I present a new shielding material process to compare and propose lightweight potential shielding fabric manufacturing processes by verifying the relationship between nanofiber patterns and shielding effectiveness.

## 2. Materials and Methods

The shielding effect of medical radiation suggests a method for attenuating the intensity of a beam while passing through a shielding body, which follows the same rule as in [21]:

$$I = I_0 \times e^{-\mu x}, \tag{1}$$

$$I = B \times I_0 \times e^{-\mu x}, \tag{2}$$

where $I_0$ and $I$ are the incident intensities of the initial and permeated beams, and $\mu$ is the linear attenuation factor. Therefore, the strength obtained by permeating the shielding body at the thickness $x$ is formed. Here, while passing through the thickness $x$, it is affected by the density $\rho$ of the shielding body and can be explained in Equation (2) using the

mass absorption coefficient $(u/p)$. It depends on the material and composition inside the shielding body generated via the incident radiation interaction [22–24]. Accordingly, the transmission strength is lost depending on the shielding material, and the attenuation of incident energy, that is, the shielding effect, can increase as the number of internal materials increases.

The spinning solution used in this study was polyurethane and tungsten powder.

Tungsten powder (W, 99.9% purity, <4 μm particle size; NanGong XinDun Alloys Spraying Co. Ltd., Xingtai, China) and polyurethane (PU, P-7195A, molecular weight 100,000–150,000; Songwon, Seoul, Republic of Korea) were utilized. Both materials were dried in an oven at 60 °C for 24 h [25].

N-dimethylformamide (DMF, 99.5%, Daejung, Siheung-si, Republic of Korea) was used as a solvent for polymer dissolution. Thus, two solvents were used to prepare the spinning solution, and chloroform (95%, Duksan, Ansan-si, Republic of Korea) was used as a powerful solvent to control the volatilization rate of the solvent and DMF for polymer dissolution. The spinning solution was mixed at 600 rpm using a magnetic stirrer (Laboratory stirrer/hot plate, PC-420, Corning, Reynosa, Mexico) after dispersing for 1 min with an ultrasonic mill after adding 5.165 g of DMF and 2.785 g of chloroform. In addition, 2.05 g of PU was added, and the speed was reduced to 220 rpm after 10 min. Subsequently, the mixture was mixed for more than 12 h to completely dissolve the polymer and then spun. To make a nanofiber pattern using the prepared spinning solution, electrospinning was implemented, as shown in Figure 1.

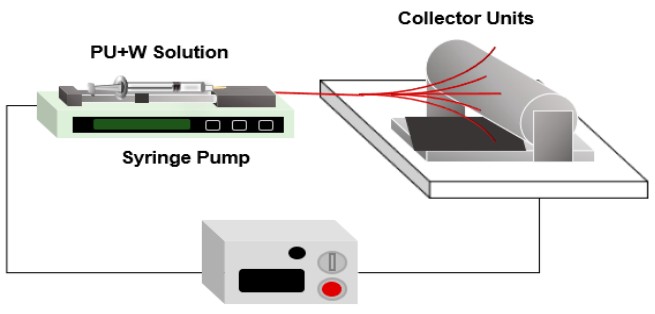

**Figure 1.** Formation of nanofiber-based shielding body.

The electrospinning conditions of the mixed spinning solution were selected using a 23–24-gauge needle and a voltage of 10 kV. The average distance of the collection plate was 13–15 cm, relative humidity was 25%–40%, and temperature was 22–25 °C. Because the pattern is formed based on the difference in temperature, distance of the collector plate, and injection time, conditions were adjusted according to the shape. In addition, it was necessary to adjust the time according to the specific gravity of tungsten. The pattern shape was repeatedly formed and well-formed when the radiation duration was reduced to less than 10 h on average. A total of 10 mL was emitted at intervals of 1 mL per hour. The shielding material fabricated with different nanofiber patterns was observed using an optical microscope (FESEM; field emission scanning electron microscope, S-4800, Hitachi, Japan) to analyze the degree of internal dispersion of the shielding material [26]. The four fabricated samples were cut into thin films using a microtome (Microtome. Leica, RM2235, Wetzlar, Germany), directly fixed on carbon paste, coated with conductive adhesive, and observed using accelerating voltages of 3 kV and 15 kV. In addition, the pattern was confirmed using a stereo microscope (Nikon Olympus, Stereo Microscope, SMZ1270, 800N, Tokyo, Japan). According to the formed pattern model, the shielding performance evaluation was set to the geometrical conditions shown in Figure 2, and the shielding rate was calculated as $1 - (W/W0) \times 100$ [27]. Here, W is the dose measured when there is a shield between the X-ray tube and the dosimeter, and W0 is calculated as the irradiation dose value measured when there is no shield between the X-ray tube and the dosimeter.

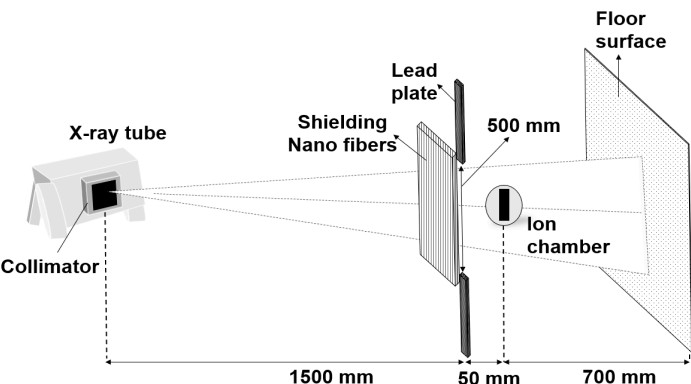

**Figure 2.** Method for evaluating shielding performance of manufactured functional fabrics.

After 10 measurements using an X-ray generator (Toshiba E7239, 150 kV-500 mA, 1999, Tokyo, Japan), the average value was calculated. The dose detector was an ion chamber (Model PM-30, 2019). The tube voltage was applied after obtaining the effective energy at 40 kVp–110 kVp, 200 mA, and 0.1 s. At this time, the correction coefficients for temperature and atmospheric pressure for accurate measurement of irradiation dose of the ionization bath dosimeter were 1.0 at a laboratory temperature of 22 °C and 1 atm, and they were used after the correction.

## 3. Results

The pattern of polymer nanofibers used as a low-dose shielding tool in medical institutions was implemented through electrospinning, as shown in Figure 3. Depending on the shape of the electrospun nanofiber pattern, the structure that can effectively disperse the metal particles showing the shielding effect can be evaluated, as shown in Figure 3. Figure 3a shows the implementation of a honeycomb shape by applying five layers in a cross-section spraying method. A pattern is formed in a matrix structure through multiple spinning; it has a five-layer shape with horizontal, vertical, and orthogonal forms (Figure 3b). Figure 3c shows the circular radiation pattern, suggesting a double-circular pattern with double radiation. The pattern shown in Figure 3d was produced using the connection method while moving the points based on the center point in the spider web spray method. The structural form of this pattern ensured that the metal particles were well dispersed in the polymer.

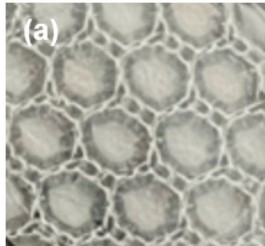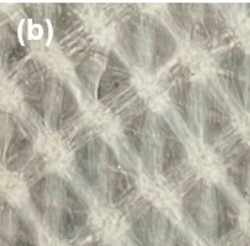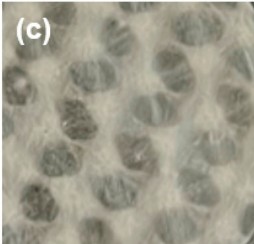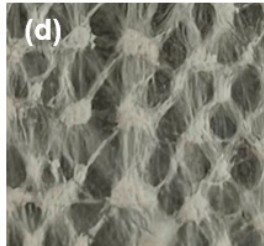

**Figure 3.** Surface image of dispersion pattern applied with polymer nanofibers (×500): (**a**) P1: honeycomb pattern; (**b**) P2: matrix pattern; (**c**) P3: double-circle pattern; (**d**) P4: spider web pattern.

Electrospinning was performed using the mixed spinning solution, and a shielding nanofiber fabric with the same thickness of 0.3 mm was realized. The cross-section of the fabricated shield was enlarged using an optical microscope to confirm the dispersion state of the metal particles. In Figure 4, the cross-section was enlarged to present the pattern and the dispersion state of the tungsten particles. As shown in Figure 4a, the tungsten particles were generally confined. In the case of the matrix pattern shown in Figure 4b, left and right orthogonal shapes were implemented; however, partial tungsten particle agglomeration

was observed. In the third case (Figure 4c), it was spun in the form of a double circle, but the tungsten particles could not be easily located because it resulted in a form in which the polymers agglomerated. Therefore, the tungsten particles were agglomerated. In Figure 4d, spinning was performed while moving the center point randomly in the form of a spider web, and some tungsten and polymer agglomeration occurred. Finally, the P1 honeycomb pattern showed the best dispersion of tungsten metal particles, while the P3 double pattern showed a large agglomeration of tungsten powder.

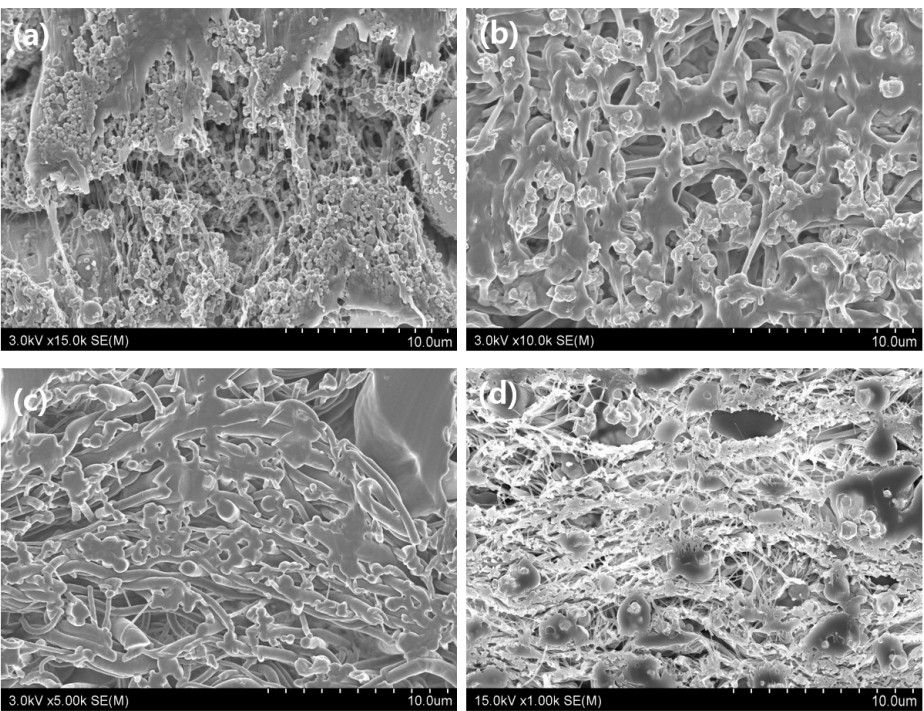

**Figure 4.** Cross-section image of radiation shielding nanofiber fabric: (**a**) P1: honeycomb pattern; (**b**) P2: matrix pattern; (**c**) P3: double-circle pattern; (**d**) P4: spider web pattern.

Therefore, the cross-section of the polymer radiation pattern was observed using an optical microscope to compare the P1 and P3 patterns, which had the best dispersion among the cross-sectional patterns of nanofibers implemented without tungsten metal particles in each pattern, as shown in Figure 5. Figure 5b shows that the more complex the pattern, the more entangled the polymers. Therefore, it is necessary to create a space in which the shielding material can be dispersed.

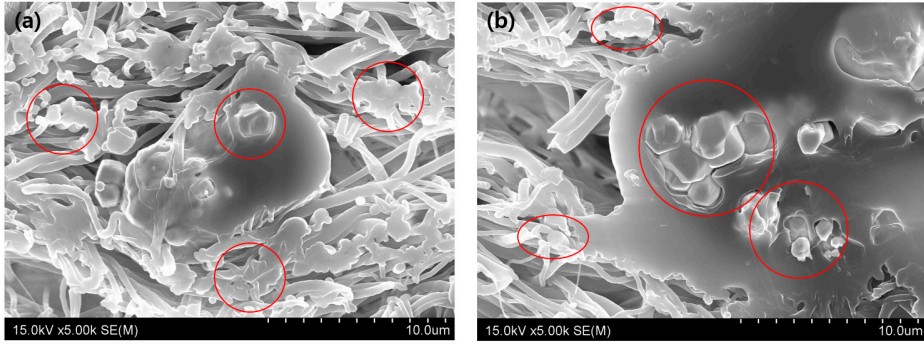

**Figure 5.** Cross-sectional image of electrospinning of nanofibers: (**a**) P1, honeycomb pattern, (**b**) P3, double-circle pattern; tungsten particle distribution in the red circles.

The shielding performance based on the fabric pattern of the nanofiber produced by electrospinning was evaluated. The results are shown in Table 1. The honeycomb pattern

of P1, in which tungsten metal particles were well dispersed, had an advantage in the shielding performance, and in the case of P3, in which polymer aggregation and tungsten powder were not evenly dispersed, the shielding performance was 26.7% lower. In the cases of P2 and P4, almost similar shielding performances were obtained. Compared with the lead equivalent of 0.25 mm, it shows a difference of approximately 8.7% with the best P1, suggesting that the shielding effect of the 0.3 mm fabric made of nanofibers is sufficient. The results of comparative analysis of the shielding performance of the 0.3 mm honeycomb pattern and the 0.25 mm lead plate are shown in Table 2. In the low incident energy region, there is a difference in shielding rate, but in the high-energy region, it is almost the same. The result is assumed to suppress the re-transmittance of the scattered photons.

**Table 1.** Evaluation of shielding performance by nanofiber electrospinning pattern.

| | Transmission Dose | Effective X-ray Energy | | | | | | | |
| | | 29.2 keV | | 34.5 keV | | 52.8 keV | | 60.3 keV | |
| | | Non | N-Fibers | Non | N-Fibers | Non | N-Fibers | Non | N-Fibers |
|---|---|---|---|---|---|---|---|---|---|
| P1 | Dose (mSv) | 0.295 | 0.0141 | 0.854 | 0.0521 | 1.117 | 0.1005 | 1.583 | 0.1708 |
| | Shielding rate (%) | 95.22 | | 93.90 | | 91.00 | | 89.21 | |
| P2 | Dose (mSv) | 0.295 | 0.0318 | 0.854 | 0.1175 | 1.117 | 0.2456 | 1.583 | 0.4686 |
| | Shielding rate (%) | 89.22 | | 86.24 | | 78.01 | | 70.40 | |
| P3 | Dose (mSv) | 0.295 | 0.0554 | 0.854 | 0.2298 | 1.117 | 0.3670 | 1.588 | 0.5928 |
| | Shielding rate (%) | 81.22 | | 73.11 | | 67.14 | | 62.55 | |
| P4 | Dose (mSv) | 0.295 | 0.0342 | 0.854 | 0.1185 | 1.117 | 0.2546 | 1.583 | 0.4874 |
| | Shielding rate (%) | 88.41 | | 86.12 | | 77.21 | | 69.21 | |

**Table 2.** Evaluation of shielding performance compared to lead plate and honeycomb pattern nanofiber fabric.

| Radiation Type | Effective X-ray Energy (keV) | Mean of Exposure (mSv) | | | Shielding Rate (%) | |
| | | Nothing | 0.25 mm Pb | P1 (0.3 mm) | 0.25 mm Pb | P1 (0.3 mm) |
|---|---|---|---|---|---|---|
| X-ray | 29.2 | 0.312 | 0.0022 | 0.0149 | 99.29 | 95.22 |
| | 34.5 | 0.854 | 0.0418 | 0.0521 | 95.11 | 93.90 |
| | 52.8 | 1.212 | 0.0806 | 0.1091 | 93.35 | 91.00 |
| | 60.3 | 1.583 | 0.1539 | 0.1866 | 90.28 | 89.21 |

For X-ray generation energy, 40 kVp~110 kVp, 200 mA, 0.1 s.

## 4. Discussion

Radiation-shielding materials directly affect the range of activities performed by medical personnel; however, the basic function of shielding should be considered. Recently, eco-friendly shielding materials have attracted attention in the field of radiation defense; thus, their effective shielding efficiency compared with that of lead must be verified [28,29]. Therefore, for functional shielding, a lightweight shielding material that can be used under lightweight conditions must be developed. Radiation protection products made of light materials, which can be always worn to prevent the scattering of rays, are considered to have a significant effect. Automatic exposure control (AEC) is also applied to the X-ray inspection equipment used by medical institutions [30].

This AEC mode forms an image by adjusting the dose according to the thickness of the area to be examined under optimal conditions; however, the dose must be increased to make the concentration difference uniform in the actual image [31–33]. Lead gloves cause a change in the dose during examination. Therefore, lighter shields should be used to reduce the incident radiation dose. As the thickness could be adjusted under the lightweight conditions used in this study, this can support the production of gloves for interventional procedures. The exposure of workers in medical institutions is mainly due to scattered rays, except when manipulating catheters by directly putting their hands in the X-ray irradiation

range [34,35]. Therefore, it is necessary to develop a shielding fabric from a lightweight and eco-friendly material that can be always worn rather than a thick lead plate to protect against scattering rays. The condition of shielding that can be used in medical radiation is the minimum lead equivalent of 0.25 mm Pb. In this study, I observed that these conditions could be satisfied by 0.3 mm tungsten nanofiber fabric. Previously, many studies have been conducted on shielding materials and polymer mixing methods for manufacturing medical radiation shielding under thin-film conditions [36]. However, it is difficult to control the thickness without a change in process technology. To match the performance of 0.25 mm Pb, the thickness must be set to almost 1.0 mm or more owing to the mixing of polymer materials [37]. In this study, the shielding performance was improved by studying the pattern format that influences the shielding performance in the manufacturing process of material of 0.3 mm thickness, similar to the lead equivalent. The lack of study of more patterns and the lack of comparison of the affinity of additional materials such as bismuth oxide and barium sulfate in addition to the single material of tungsten are considered as the limitations of the study. This study focused on nanofibers to achieve thin-film conditions and light weight. In previous studies, a method of separating the shielding and polymer materials was proposed; however, methods to improve the shielding performance by solving the problems of electrospinning and forming a radiation pattern have not been studied. Electrospinning patterns of nanofibers are generally irregular; however, forming a pattern similar to the one in this study can affect the shielding performance of the fabricated sheet [38]. The pattern shape is formed based on the electrospinning conditions, and the formed pattern was intended to form a shape that can store metal particles [39]. Therefore, this study, for the first time, proposes that shielding performance can be improved by changing the pattern according to electrospinning. The experiment demonstrated that a shielding fiber fabric almost similar to the lead equivalent can be manufactured. In particular, it is a very important task to disperse the shielding material for nanofibers that can easily achieve thin-film conditions [40]. Although there are many existing studies, it was revealed that the shielding performance can be sufficiently adjusted by the pattern shape. For example, the honeycomb structure, which is a structure that can effectively contain the shielding material, was more effective than a complex pattern. In the future, if a wearable product with a lightweight radiation protection function is needed, weaving will be possible through nanofiber-based functional radiation protection fabric.

## 5. Conclusions

A lightweight and eco-friendly shielding material based on nanofibers was designed to develop a functional radiation protection fabric. It was mainly developed for protection against scattering rays and for the production of wearable fabrics. To improve the shielding performance, honeycomb, matrix, bull, and spider web patterns were produced via electrospinning using a mixed solution of polyurethane and tungsten. The honeycomb pattern exhibited excellent shielding performance, and the double-circle pattern exhibited considerably poorer shielding performance; there was a difference of 26.7% in the shielding performance between the two patterns. The shielding performance was approximately 8.7% lower than that of 0.25 mm of Pb, suggesting sufficient potential. Consequently, the electrospinning pattern is a very important condition for manufacturing nanofiber-based shielding fabrics.

**Funding:** This research was supported by Basic Science Research Program through the National Research Foundation of Korea (NRF) funded by the Ministry of Education (2020R1I1A3070451).

**Institutional Review Board Statement:** Not applicable.

**Informed Consent Statement:** Not applicable.

**Data Availability Statement:** Not applicable.

**Conflicts of Interest:** The author declares no conflict of interest.

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
