# Peer review of "Study on the Changes in Shielding Performance Based on Electrospinning Pattern Shapes in the Manufacturing Process of Polymer-Metal Composite Radiation Shielding Materials"

_coatings, doi:10.3390/coatings13061028_

Round 1
Reviewer 1 Report
Dear Author,
The author is trying to investigate the shielding performance of new Polymer-Metal Composite Radiation Shielding Materials.
Abstract:
1. The abstract is written in more generalized form. Try avoiding phrases “ I solved the problem “
2. Expand abbreviation Pb.
3. Write the conclusion of the study
Introduction :
1. The author is suggested to explain more about the “metal particle dispersion method”
Discussion :
The author can enhance the discussion by incorporating similar studies and making a comparison with other studies.
Limitation of the study needs to be mentioned.
Minor spell checks are required .
Reviewer 2 Report
Effective protection against X-ray is the premise of utilizing the X-ray, thus it is critical to develop novel X-ray shielding materials with both low density and high X-ray attenuation efficiency. This issue is especially relevant for medical professionals involved in a number of procedures, such as: Fluoroscopy, CT Scans, Mammography, Dental X-rays, Treating Cancer With X-ray Radiation Therapy, etc. Reducing the weight of the shield worn by medical personnel in the radiation generating area plays a key role in improving their productivity and mobility. Along with the traditional shielding material – lead, today often pay attention to materials such as tungsten, bismuth oxide, barium sulfate and boron. Using such materials, it is possible to create various types of protective equipment, for example, plates or protective panels. In this article, the author develops an effective fiber-type shield. A fiber-type shield is woven from a yarn impregnated with the shielding material. However, the shielding performance is limited by the pinholes generated between the yarn during the weaving process. Therefore, on the practical application of this study, I believe that fiber-based screens can only have a significant effect on protection against secondary (or scattered) radiation.
The idea of the study is interesting and deserves attention. In this article, the author developed new shielding material process to compare and propose lightweight potential shielding fabric manufacturing processes by verifying the relationship between nanofiber patterns and shielding effectiveness. This article is the development of a whole line of research, the beginning of which the author initiated in his previous publications. The article is rather concise, but the study was carefully conducted, and its results are clear and understandable. The research topic is very relevant. The article has a scientific novelty, and the implementation of its results will be of practical value. I believe that this article can be published as presented.

Reviewer 3 Report
The work concerns the study of materials composed in such a way as to protect people against ionizing radiation while allowing them to maintain as much freedom of movement as possible. And this is the most valuable part of the manuscript. However, improvement should be made in particular with regard to the part concerning the method and results of measuring radiation attenuation.
The attenuation efficiency tests were performed for several low-energy lines. The author should explain why this range was chosen. Moreover if the doses are being assessed, exposure time should also be determined. Otherwise, this information is of little use. Therefore, the author should clarify a few things to make the article more understandable for the reader.
- Formula 1. Instead of quoting the reference [22], it would be better to point out that this is Beer-Lambert law for monoenergetic radiation
- Formula 2. How did you apply this formula in your investigations ? I did not find such relationship both in the reference [22] and reference [23]. Moreover the reference [23] corresponds to neutron radiation. Sometimes instead of the linear coefficient is used the mass attenuation coefficient that is equal to linear coefficient divided by material density, and then the thickness should be replaced by the surface mass of material. Maybe that's the point, but in that case it should be worded more clearly.
- I did not find a reference to table 2 in the manuscript. Please introduced the exposure time that was used to evalute doses
- Line 127: humidity was 25-40 -> relative humidity was 25-40 %
Round 2
Reviewer 3 Report
Thank you for the revised version. I have only two remarks.
I meant that in the manuscript there should be a sentence referring to table 2. , for example "In table 2 ....." or "The shielding performance are presented in Table 2". lt should be somehow mentioned in the manuscript. Something like that is still missing.
In my opinion It would be better, for the sake of the readers, to introduce the exposition time (0.1 s) also in Table2.
